# Biomimetic Calcium Phosphate Coatings for Bioactivation of Titanium Implant Surfaces: Methodological Approach and In Vitro Evaluation of Biocompatibility

**DOI:** 10.3390/ma14133516

**Published:** 2021-06-24

**Authors:** Thomas Kreller, Franziska Sahm, Rainer Bader, Aldo R. Boccaccini, Anika Jonitz-Heincke, Rainer Detsch

**Affiliations:** 1Department of Materials Science and Engineering, Institute of Biomaterials, Friedrich Alexander-University Erlangen-Nuremberg, 91058 Erlangen, Germany; thomas.kreller@fau.de (T.K.); aldo.boccaccini@fau.de (A.R.B.); 2Research Laboratory for Biomechanics and Implant Technology, Department of Orthopedics, Rostock University Medical Center, 18057 Rostock, Germany; Franziska.Sahm@med.uni-rostock.de (F.S.); Rainer.Bader@med.uni-rostock.de (R.B.); Anika.Jonitz-Heincke@med.uni-rostock.de (A.J.-H.)

**Keywords:** hydroxyapatite, interleukins, primary human osteoblasts, Ti6Al4V

## Abstract

Ti6Al4V as a common implant material features good mechanical properties and corrosion resistance. However, untreated, it lacks bioactivity. In contrast, coatings with calcium phosphates (CaP) were shown to improve cell–material interactions in bone tissue engineering. Therefore, this work aimed to investigate how to tailor biomimetic CaP coatings on Ti6Al4V substrates using modified biomimetic calcium phosphate (BCP) coating solutions. Furthermore, the impact of substrate immersion in a 1 M alkaline CaCl_2_ solution (pH = 10) on subsequent CaP coating formation was examined. CaP coatings were characterized via scanning electron microscopy, x-ray diffraction, energy-dispersive x-ray spectroscopy, and laser-scanning microscope. Biocompatibility of coatings was carried out with primary human osteoblasts analyzing cell morphology, proliferation, collagen type 1, and interleukin 6 and 8 release. Results indicate a successful formation of low crystalline hydroxyapatite (HA) on top of every sample after immersion in each BCP coating solution after 14 days. Furthermore, HA coating promoted cell proliferation and reduced the concentration of interleukins compared to the uncoated surface, assuming increased biocompatibility.

## 1. Introduction

Implant materials have to meet versatile demands. They need to feature sufficient mechanical properties, corrosion resistance, and good biocompatibility [1]. Among the metals, titanium and its alloys are considered as the gold standard due to their biocompatibility, corrosion resistance, and their lack of harmful elements such as nickel, cobalt, and chromium [2]. However, the mechanical properties of commercially pure titanium (cp-Ti) are not sufficient for high mechanically loaded implants nor can they withstand intensive wear. To resolve these limitations, cp-Ti was substituted by α + β-type Ti-based alloys (i.e., Ti6Al4V) [3]. Ti6Al4V (90 wt % titanium (Ti), 6 wt % aluminum (Al), and 4 wt % vanadium (V)) is often used as an implant material in dental applications or orthopedic surgery because of its above-mentioned favorable properties and its corrosion-resistant titanium oxide layer [3,4,5,6]. However, dense titanium and its alloys can be encapsulated by fibrous tissue after implantation due to limited ingrowth of bone and weak interfacial bonds to the surrounding tissue [3,7,8]. Consequently, the goal is to functionalize the surface of Ti6Al4V substrates since the surface characteristics of implants influence biological responses at the interface between implant material and biological surroundings [9].

In this context, surface functionalization of Ti6Al4V with calcium phosphate (CaP) coatings are of great interest by providing enhanced osteoconductive properties [10,11,12,13,14,15]. CaP is further known to improve proliferation and differentiation of osteoblast-like cells [16,17,18] and to improve implant ingrowth in vivo [19,20]. Special cytokine levels can be measured after a successful or unsuccessful integration of the implant. Vascular endothelial growth factor (VEGF) secreted from different cell types, like fibroblasts, bone marrow stromal cells, osteoblasts as well as osteoclasts are known a so-called “survival factor”, suggesting reduced or in some cell types enhanced VEGF production [21]. An increase of interleukin (IL-)6 and IL-8 is known to be related with a higher implant failure rate and deteriorated bone formation [22,23,24,25,26]. A reduction in IL-6 and IL-8 can lead to an enhanced osteogenic differentiation, but IL-6 is also known to improve osteogenic differentiation in stem cells [27,28,29]. However, CaP coatings with their promising biological responses can influence the ingrowth and the failure rate of implants. Apart from these, CaP coatings also have interesting mechanical and topographical aspects. As a consequence of the improved osteoconductivity of CaP coatings, a stable interface between bone and implant can be established in cementless fixation of implants [30] and the friction coefficient between the implant and surrounding tissue can be reduced, resulting in a good implant fixation [31]. Furthermore, as presented in this work, due to the adjustment of the coating technique as well as coating parameters, it is possible to tailor surface characteristics like surface roughness and coating composition. Common coating techniques involve plasma spraying, electrophoretic deposition, sputter deposition, and sol-gel techniques [32]. In this work, however, chemical treatments were used to enable the induction of bioactive surface characteristics to the titanium alloy substrates. These include acid etching in hydrochloric acid (HCl) for micro-structuring and a subsequent sodium hydroxide (NaOH) treatment to form a nano-structured, bioactive sodium titanate layer on the surface [33]. Subsequently, the modified surface acts as an origin for the in vitro-nucleation of CaP from biomimetic CaP (BCP) coating solutions derived from simulated body fluid (SBF) [8,33,34,35]. Despite long immersion times in BCP coating solutions, the hereby presented biomimetic approach represents a cost-effective and low-temperature process generating bonelike nanostructured apatite crystals on Ti6Al4V featuring high bioactivity and anti-inflammatory characteristics. Although SBF mimics the ion concentration and pH of human blood plasma at physiological temperatures, modifications like supersaturations are advisable to improve the coating process [36]. It was shown that CaP formation could be accelerated using supersaturated SBFs simplifying the initial nucleation [7,32,37,38]. However, due to the focus on layer formation kinetics, the effects of the supersaturation degree on surface topography, surface chemistry, and the resulting biological cell response remain often underrepresented. Therefore, our present work aimed to evaluate and compare the structure, morphology, composition, homogeneity, and biocompatibility of CaP-coatings on Ti6Al4V surfaces induced by a chemical pre-treatment and subsequent immersion in variously supersaturated BCP coating solutions (Table 1). Although these coating solutions were derived from SBF as presented by Kokubo et al. [7], they are intended to be explicitly adapted for coating Ti6Al4V substrates with CaP and not to be understood as test solutions for bioactivity assays. For this reason, we refer to BCP coating solutions in this work and refrain from using the term SBF. Moreover, the impact of immersing the Ti6Al4V substrates in 1 M alkaline calcium chloride (CaCl_2_; pH = 10) solution for 24 h before immersion in supersaturated BCP coating solution is examined. Doing this, we hypothesize that sodium ions are substituted with calcium ions in the hydrogen titanate layer on the titanium alloy substrates, which could additionally affect the CaP formation during subsequent immersion in a BCP coating solution. The biological response of human primary osteoblasts on biomimetic CaP coatings was assessed via DNA isolation, collagen I synthesis, and interleukin as well as vascular endothelial growth factor release, allowing conclusions about the anti-inflammatory behavior and the successful integration as a bone implant.

## 2. Materials and Methods

### 2.1. Biomimetic CaP-Coated Ti6Al4V Substrates

Ti6Al4V plates (diameter: 15 mm, thickness: 1 mm, corundum blasted, R_z_ = 20 µm, DOT, Rostock, Germany) were microroughened via immersion in 37% HCl for 2 h at 50 °C, washed with ultrapure water (UPW), and soaked in 10 M NaOH aqueous solution at 60 °C for 24 h. Subsequently, the substrates were rinsed with UPW before further processing. Biomimetic CaP coatings were generated via immersion in modified BCP coating solution at 37 °C for 14 days (Appendix A). BCP coating solutions were prepared similarly as described by Kokubo et al. [7], however, with modified ion concentrations (Table 1, Appendix A). Briefly, sodium chloride (NaCl), potassium chloride (KCl), sodium hydrogen carbonate (NaHCO_3_), di-potassium hydrogen phosphate trihydrate (K_2_HPO_4_·3H_2_O), magnesium chloride hexahydrate (MgCl_2_·6H_2_O), calcium chloride dihydrate (CaCl_2_·2H_2_O), and sodium sulfate (Na_2_SO_4_) were dissolved in UPW. The pH was adjusted to pH = 7.4 at 37 °C using Tris-hydroxymethyl aminomethane (Tris) and 1 M HCl. Besides, chemically pre-treated Ti6Al4V plates were soaked in a 1 M CaCl_2_ solution (pH = 10) for 24 h at 37 °C before immersion in BCP coating solution. Substrates were directly transferred into BCPx1.5 for 14 days without additional washing steps. CaCl_2_ solution’s pH value was adjusted to pH = 10 via the addition of NaOH. BCP coating solutions were exchanged every second day.

### 2.2. Surface Analysis

Six Ti6Al4V plates were prepared of each sample type (Ti6Al4V, BCPx1, BCPx1.5, BCPx2, and CaBCPx1.5) for surface analysis. Surface topography was analyzed via a scanning electron microscope (*SEM*, Auriga CrossBeam, Carl Zeiss Microscopy GmbH, Oberkochen, Germany). Ti6Al4V substrates were rinsed with UPW and dried at room temperature (RT = 22 °C) for 24 h before SEM analysis. Structural identification of CaP-layers was conducted via x-ray diffraction (XRD, MiniFlex600, Rigaku, Japan). CaP-coated Ti6Al4V samples were dried for 24 h at room temperature before diffractograms were recorded. Crystalline phases were determined using Cu Kα radiation at a scan rate of 4 min^−1^ over a 2θ range of 5–90°. The crystallinity of the CaP coatings was determined using Equation (1) (Appendix A). A_HA_ was the integrated area of HA peaks at 2θ = 26° and 2θ = 32° of HA powder. A_C_ was the integrated area of HA peaks at 2θ = 26° and 2θ = 32° of CaP coatings. The crystallite size of HCA was calculated from the (0 0 2) peak at 2θ = 26° using the Scherrer equation (Equation (2), Appendix A). K was the Scherrer’s constant, λ was the wavelength of the x-ray source, β was the FWHM (full width at half maximum) in radians, and θ was the peak position in radians. Arithmetic average roughness (R_a_) and 3D microstructural surface profile scans were conducted with a laser scanning microscope (LSM, Lext OLS 4000 Laser Microscope, Olympus, Hamburg, Germany). Dried samples were used for 3D Scanning and R_a_ determination. Microstructural surface scans covered a surface area of 4 mm^2^. Average R_a_ values consist of *n* = 15 randomly distributed line roughness measurements. Each line roughness measurement covered a length of l = 2 mm. Elemental analysis was performed using energy-dispersive x-ray spectroscopy (EDX) at 20.0 kV using the Auriga CrossBeam SEM (Carl Zeiss Microscopy GmbH, Oberkochen, Germany).

### 2.3. Cell Culture of Primary Human Osteoblasts

Primary human osteoblasts were isolated as described previously [39]. In brief, femoral heads were collected from patients undergoing a primary total hip replacement under sterile conditions with the consent of the patients, following approval by the Local Ethical Committee (Registration number: A 2010-0010, approval date: 27 January 2017). Cancellous bone was isolated out of the femoral heads and digested with collagenase and dispase (Roche, Basel, Switzerland). After different filtration and centrifugation steps, the remaining cells were transferred in a cell culture flask and cultivated in Dulbecco’s Modified Eagle Medium (DMEM, PAN-Biotech, Aidenbach, Germany), containing 10% fetal calf serum (FCS, PAN-Biotech, Aidenbach, Germany), 1% amphotericin B, 1% penicillin-streptomycin, and 1% Hepes buffer (all: Sigma-Aldrich, Munich, Germany). Cultivation was done under standard cell culture conditions (5% CO_2_ and 37 °C) for two passages. Ascorbic acid (final concentration: 50 μg/mL), β-glycerophosphate (final concentration: 10 mM), and dexamethasone (final concentration: 100 nM) (all: Sigma-Aldrich, Munich, Germany) were added to the cell culture medium to promote osteogenic differentiation. The human osteoblasts were stored in liquid nitrogen until usage. For the experiments, cells from five different donors, three females (age: 75 ± 10 years) and two males (age: 60 ± 7 years), were thawed, cultured for another passage, and then seeded on six CaP-coated Ti6Al4V and six untreated Ti6Al4V slides with a density of 22,637 cells/cm^2^ in a 12 well plate (passage 4). Cells adhered 30 min at room temperature on the slides and were cultured in 2 mL of cell culture medium containing osteogenic additives over 48 h.

### 2.4. Cell Proliferation

The cell proliferation was quantified through the DNA content using the peqGOLD Tissue DNA Mini Kit (peqlab, Erlangen, Germany) and the Quant-iT™ PicoGreen™ dsDNA Assay Kit (Life Technologies Corporation, Eugene, OR, USA). Initially, cells were lysed for 30 min with the DNA Lysis Buffer T and isolated as described in the user manual. DNA was eluted from the DNA binding columns with 20 µL elution buffer. For quantification, eluted DNA was diluted 1:20 in 1 × TE and mixed afterward 1:2 with the pico green reagent as described in the manufacturer’s instructions. The fluorescence was measured with the Infinite F200 pro (Tecan Trading AG, Maennedorf, Switzerland) using an excitation of 485 nm and an emission of 535 nm.

### 2.5. Cell Morphology

Observations on cell morphology were done with the scanning electron microscope (Auriga CrossBeam, Carl Zeiss, Oberkochen Germany) and the field emission scanning electron microscope (MERLIN VP Compact, Carl Zeiss, Oberkochen Germany) from the selected regions on the slides. Therefore, the medium was aspirated and the slides were washed twice with phosphate-buffered saline (Sigma-Aldrich, Munich, Germany) incubated under cell culture conditions for 5 min each time. Afterward, the cells were fixed in fixation buffer (1% paraformaldehyde, 2.5% glutaraldehyde, 0.1 M sodium phosphate buffer, pH 7.3) and stored at 4 °C. The samples were dehydrated with an ethanol acetone series and critical point dried using CO_2_ (Leica EM CPD 300, Leica Microsystems, Germany).

### 2.6. Quantification of Procollagen Type I Synthesis

The type I C-terminal collagen propeptide (CICP) was used as a marker for the collagen I synthesis of osteoblasts. Protein contents in supernatants were analyzed using the MicroVue CICP ELISA (Quidel, San Diego, CA, USA) according to the manufacturer’s instructions. Internal standards served for the determination of the CICP concentration. The absorption was measured using a microplate reader (Tecan Trading AG, Maennedorf, Switzerland) at a wavelength of 405 nm. The measured CICP concentration from each sample was normalized using the corresponding total protein content of the supernatant.

### 2.7. Quantification of Inflammatory Signals

The concentrations of interleukin (IL-)6 and IL-8 were determined in supernatants of osteoblastic cell cultures. The IL-6 Human uncoated ELISA Kit and the IL-8 Human uncoated ELISA Kit (both: Thermo Fisher Scientific, Waltham, MA, USA) were implemented as described in the manufacturer’s instructions. The absorption was measured using an Infinite F200 pro (Tecan Trading AG, Maennedorf, Switzerland) at a wavelength of 405 nm. The measured IL-6 and IL-8 concentrations were normalized using the corresponding total protein content of the supernatant.

### 2.8. VEGF

The concentrations of VEGF were determined in the supernatants of the osteoblast cell culture medium. The ELISA assay (RayBiotech, Peachtree Corners, Georgia, USA) was carried out according to the protocol supplied by the manufacturer and measured spectrometrically at 450 nm using a microplate reader (FLUOstar Omega, BMG Labtech, Germany). The measured VEGF concentrations were normalized using the corresponding total protein content of the supernatant.

### 2.9. Total Protein Content

The total protein content of the supernatants was measured using the Qubit fluorometer Q32857 and the Invitrogen Qubit Protein Assay Kit (both: Thermo Fisher Scientific, Waltham, MA, USA). The assay was carried out following the manufacturer’s instructions. Included standards were used to quantify the total protein content.

### 2.10. Statistical Analysis

Data are expressed as mean ± standard deviation (SD). Statistical analysis was performed using Origin2020 software (OriginLab Corporation, Northhampton, Massachusetts, USA) or the GraphPad Prism software (GraphPad Software, San Diego, California, USA). All experiments regarding the structure and composition of CaP-coatings were performed using a minimum of *n* = 3 replicates. Biocompatibility tests used *n* = 5 replicates. Normality tests and analysis of variance homogeneity were performed using the Shapiro–Wilk test. Statistically significant differences between means were determined at a value of *p* < 0.05 as determined by the Tukey post-hoc test using a one-way analysis of variances (ANOVA) for normal distribution and the Friedmann analysis for not normal distribution. Different significance levels (*p*-values) are indicated with asterisks or hashes and the specific *p*-value is provided in each figure legend.

## 3. Results

### 3.1. Structure and Composition of CaP-Coatings

Scanning electron microscopy revealed biomimetically altered surface topographies due to the surface functionalization of Ti6Al4V substrates and subsequent immersion in modified BCP coating solutions for 14 days (Figure 1A). Chemically untreated Ti6Al4V surfaces without immersion in BCP coating solution served as a reference. The degree of BCP coating solution supersaturation (x1, x1.5, and x2) could directly be correlated with surface topography and microstructure (Figure 1A). BCPx1 induced the formation of heterogeneous distributed cauliflower-like CaP structures indicating the formation of hydroxyapatite (HA) [7,33,40,41,42,43]. BCPx1.5 revealed similar CaP structures. However, they covered the complete substrate surface and created a homogenous coating.

CaBCPx1.5 and BCPx2 featured heterogeneous CaP surface structures that covered the entire substrate surface. CaCl_2_ pre-treatment changed the CaP microstructure. Surface morphologies were confirmed with a qualitative roughness assessment via 3D surface profilometry (Figure 1B). Ti6Al4V, BCPx1, and BCPx1.5 featured smooth microstructural roughness profiles. The initial waviness of the surface profile can be traced back to the corundum blasting of the Ti6Al4V substrate. BCPx1.5 featured the smoothest appearing surface. Complementary to SEM observations, CaBCPx1.5 and BCPx2 substrates appeared rough and heterogeneous. The observations were substantiated with arithmetic average roughness (R_a_) determinations conducting line roughness measurements (Figure 2D). The arithmetic average roughness (R_a_) of the untreated Ti6Al4V substrate (R_a_ = 3.40 ± 0.17 µm) and BCPx1 immersed substrates (R_a_ = 3.88 ± 0.39 µm) featured higher R_a_ values compared to BCPx1.5 (R_a_ = 2.50 ± 0.16 µm) (Figure 2D). BCPx2 (R_a_ = 11.29 ± 1.50 µm) and CaBCPx1.5 (R_a_ = 14.25 ± 1.49 µm) featured the highest surface roughness. XRD analysis identified the precipitated CaP as low crystalline hydroxyapatite for all BCPs. The almost amorphous HA layers were visualized in x-ray diffractograms (Figure 2C) by the presence of weak and broad hydroxyl carbonated apatite (HCA) related peaks at 2θ = 25.78° (0 0 2), 2θ = 28.9° (2 1 0), and 2θ = 31.74° (2 1 1) [44,45,46]. Using Scherrer’s equation, decreasing crystallite sizes were observed for increasing BCP coating solution supersaturations (Figure 2A). The CaCl_2_ pre-treatment increased the crystallite size compared to BCPx1.5 from 16.74 nm to 30.04 nm. Crystallinity behaved the other way around. Here, the crystallinity increased from 10.94% (BCPx1) to 43.58% (BCPx2) due to an increased BCP coating solution supersaturation (Figure 2A). CaBCPx1.5 (15.15%) led to a decrease of crystallinity compared to BCPx1.5 (25.92%). EDX analysis detected substrate contents (titanium (Ti), aluminum (Al), vanadium (V)) as well as biomimetic coating elements (calcium (Ca), phosphorus (P), magnesium (Mg), sodium (Na), carbon (C), and oxygen (O)) (Figure 2A,B). The quantitative elemental analysis detected that BCPx1 (Ca = 9.95 at%, *p* = 6.54 at%) generated lower Ca and P contents compared to BCPx1.5 (Ca = 15.66 at%, *p* = 10.44 at%) and BCPx2 (Ca = 17.21 at%, *p* = 10.32 at%). However, Ca and P contents were decreased when comparing CaBCPx1.5 (Ca = 13.32 at%, *p* = 8.68 at%) to BCPx1.5. Consequently, derived Ca/P ratios varied depending on the respective BCP coating solution’s ion concentration and CaCl_2_ pre-treatment. BCPx1 treated substrates featured a Ca/P ratio of Ca/P = 1.52, BCPx1.5 of Ca/P = 1.5, CaBCPx1.5 of Ca/P = 1.53, and BCPx2 of Ca/P = 1.67. Substrate elements (Ti, V) could be detected in the BCPx1 and CaBCPx1.5 samples despite the CaP coating.

### 3.2. In Vitro Biocompatibility of CaP-Coatings

Analyzing the in vitro biocompatibility of the different CaP-coatings, human osteoblasts were seeded on the coatings as well as on the pure Ti6Al4V. The DNA content served as a marker for cell proliferation. Due to the variability of the different primary donor cells, high standard deviations occurred, leading to non-significant changes in the DNA concentration. However, a lower number of cells on the Ti6Al4V slides without coating was determined (Figure 3A). All coatings led to a higher DNA concentration with a slightly higher DNA content for BCPx1 and a slightly lower DNA content for BCPx2.

The amount of type I C-terminal collagen propeptide of osteoblasts seeded on the different surface modifications did not differ significantly compared to Ti6Al4V (Figure 3B). Juxtaposing the different surface modifications among each other, a significant upregulation was detected by cells growing on BCPx1.5 compared to BCPx1 (*p* = 0.0487) and CaBCPx1.5 (*p* = 0.0047). Cells cultured on CaBCPx1.5 released the lowest concentration of CICP.

The human osteoblasts adhered well to all surfaces (Figure 4). Cells attached to the different surface modifications and span over the shapes of the Ti6Al4V and the CaP structures. No significant differences on cell morphology of the CaP structures could be noticed. On all CaP surfaces, osteoblasts extensively span the structures. In contrast to the titanium surface and the BCPx2 coating, osteoblasts on BCPx1, BCPx1.5, and CaBCPx1.5 appear to form more numerous filopodia. This is also evident in the lower magnification of the overview images (Appendix A).

### 3.3. Induction of Cytokine Release by Human Osteoblasts

Release of IL-6, IL-8, and VEGF in supernatants of the different cell-seeded surfaces was determined to evaluate the induction of inflammatory processes. The concentration dependence of IL-6 resembled that of IL-8 over the different surface modifications. Cells grown on Ti6Al4V produced slightly higher concentrations of IL-6 as well as IL-8 compared to the BCP coatings. Only BCPx1 showed a similar IL-6 and IL-8 concentration to Ti6Al4V (Figure 5). The median amount of IL-6 from cells cultured on BCPx1.5 was slightly lower than from cells cultured on BCPx1. Surface modifications with CaBCPx1.5 and BCPx2 significantly reduced (*p* = 0.250 for CaBCPx1.5 and *p* = 0.0333 for BCPx2) the average released IL-6 concentration (Figure 5A). Focusing on IL-8, BCPx1 had the highest average IL-8 concentration of all CaP coatings, significantly higher than CaBCPx1.5 (*p* = 0.0114) and BCPx2 (*p* = 0.0092). Cells seeded on BCPx1.5 (*p* = 0.0037), CaBCPx1.5 (*p* = 0.0006), and BCPx2 (*p* = 0.0005) released significantly diminished amounts of IL-8 (Figure 5B). The highest VEGF concentrations were measured for Ti6Al4V, CaBCPx1.5, and BCPx2. Coatings with BCPx1 and BCPx1.5 led to a reduced amount of VEGF in the supernatant. CaBCPx1.5 was significantly upregulated compared to BCPx1 (*p* = 0.0373) and BCPx1.5 (*p* = 0.0373).

## 4. Discussion

### 4.1. Structure and Composition of CaP-Coatings

Chemical pre-treatment of Ti6Al4V substrates with NaOH enabled HCA formation after exposition in BCP coating solution as described in the literature [33,47,48]. Compared to Yamaguchi and Kokubo et al., no heat-treatment was carried out after substrate exposure in NaOH, consequently, the sodium hydrogen titanate layer will not be dehydrated and densified into sodium titanate [49]. This could simplify the sodium ion release due to less dense lattice structures and could therefore lead to an enhanced pH increase. Increasing pH could lead to accelerated apatite nucleation due to increased BCP coating solution supersaturations [33]. Furthermore, no heat-treatment allowed the substitution of sodium ions with calcium during the treatment in 1 M CaCl_2_ solution, as shown by Kokubo et al. [48,50]. During the substitution process, sodium hydrogen titanate was transformed into calcium hydrogen titanate [48]. SEM images revealed that all formed CaP coatings independent of CaCl_2_ pre-treatment were composed of nanosized platelets forming spherical aggregates after BCP coating solution exposure. These are characteristic for coatings precipitated from SBF [7,33,43,51,52,53]. However, using BCPx1, spherical CaP aggregates were only heterogeneously distributed over the surface in contrast to observations from Kokubo et al. [7]. This could be explained with Ostwald’s nucleation theory, in which the free energy for nucleation ΔG depends on the supersaturation of the solution (S), the net interfacial energy for nucleation (σ), the temperature (T), and the particle surface area (A): ΔG = −RTlnS + σA [38,53,54]. Since BCPx1 featured decreased Ca^2+^ ion concentrations (1.99 mM) compared to Kokubo’s formulation (2.5 mM) [7], BCP coating solution supersaturation was comparatively lower. Consequently, apatite nucleation was not as energetically favored, resulting in the heterogeneous distribution of spherical CaP aggregates. In the intermediate spaces between the spherical aggregates, a CaP structure is visible that is expected to be a precursor CaP phase to HA. Literature suggested that the hydroxyapatite precursors could have been amorphous calcium phosphates (ACPs) or octa calcium phosphates (OCPs), which can later transform into a crystalline phase [33,55,56,57]. This assumption was substantiated by the EDX analysis. Atomic Ca/P ratios were found to be 1.52 for the spherical CaP structures and 1.15 for the intermediate CaP structure. Low Ca/P ratios in this order of magnitude usually correspond to di- or octacalciumphosphates [58]. The alloy elements of the Ti6Al4V substrate (Al, V) were not expected to be the reason for the decreased apatite-forming ability of BCPx1 since Kim et al. [59] could prove that Al and V were released during the NaOH treatment, resulting in an aluminum and vanadium free sodium titanate hydrogel layer. However, other titanium alloy elements like niobium (Nb), zirconium (Zr), and tantalum (Ta) are less soluble in NaOH and can suppress the release of sodium ions during the immersion in BCP/SBF and can therefore hinder apatite-formation [60]. In parallel to BCPx1, Ostwald’s theory also explained enhanced apatite nucleation for BCPx1.5 due to increased BCP coating solution supersaturation, which led to homogeneously and matured CaP coatings. Similar microstructures were generated by Bigi et al. [61]. Ti6Al4V was pre-treated with HF and HNO_3_ and immersed in SBF, however, with increased Ca^2+^ concentrations (3.8 mM) compared to BCPx1.5. BCPx2 resulted in heterogeneous CaP structures possibly due to increased apatite nucleation and crystal growth in preferred orientations [43]. The CaCl_2_ pre-treatment seemed to further improve CaP formation. Although Ti6Al4V substrates were immersed in BCPx1.5, heterogeneous structures like in BCPx2 were formed. Most likely, this was due to the high 1 M CaCl_2_ concentration. In comparison, Bütev et al. soaked Ti6Al7Nb in a 0.1 M CaCl_2_ solution at 40 °C for 24 h after the NaOH treatment, resulting in homogeneous apatite coatings upon immersion in SBF for 15 days [62]. Consequently, this could indicate a tunable CaP formation on titanium and its alloys by adjusting the CaCl_2_ pre-treatment concentration. In general, enhanced crystal growth for BCPx1.5, CaBCPx1.5, and BCPx2 could be explained by improved mass transport across the interface between the crystal and the surrounding fluid due to supersaturation [63]. These assumptions were in accordance with the qualitative (surface profiles) and quantitative surface roughness evaluation (line roughness measurements). The arithmetic roughness (R_a_) of pure Ti6Al4V substrates was strongly dependent on mechanical treatments. An arithmetic roughness R_a_ = 3.4 ± 0.17 µm could be explained by corundum blasting and resembled literature values with similar mechanical treatments [64,65,66]. Heterogeneous CaP coating formation for BCPx1 explained minor R_a_ differences compared to the pure Ti6Al4V reference. Decreased R_a_ values of BCPx1.5 samples could be explained by enhanced crystal growth due to BCP coating solution supersaturation and preferred crystal growth orientations of HCA crystals in the direction of their c-axis [43,67]. Craters on Ti6Al4V surfaces created by corundum blasting and HCl micro structuring could therefore be smoothed by the enhanced HCA crystal growth using BCPx1.5. Line roughness measurements of CaBCPx1.5 complement SEM observations. Due to the CaCl_2_ pre-treatment, heterogeneous crystal growth was enhanced, resulting in rough surface profiles. BCPx2 supersaturation was expected to induce strongly enhanced crystal growth in preferred directions due to improved mass transport, resulting in high surface roughness. XRD analysis identified all CaP coatings as low crystalline hydroxyapatite. The highly amorphous HA layers were visualized in x-ray diffractograms by the presence of weak and broad hydroxyl carbonated apatite (HCA) related peaks at 2θ = 25.78° (0 0 2), 2θ = 28.9° (2 1 0), and 2θ = 31.74° (2 1 1) [44,45,46]. Low crystallinity could be traced back to incorporated impurities (i.e., carbonate (CO_3_^2−^), magnesium (Mg^2+^), and sodium (Na^+^) ions) and small size crystals [68]. However, it was found that amorphous coatings could be beneficial for early bone ingrowth compared to coatings with high crystallinity due to a greater similarity with natural bone tissue [69,70]. Natural bone mineral, usually calcium-deficient apatite, is characteristic for small size crystals and non-stochiometry, enabling osteoclastic bone resorption [56]. In this work, nano-sized crystals ranging from 16.64 nm to 44.06 nm were found to be similar to natural bone apatite [71]. BCP supersaturation-dependent crystal sizes might be explained via nucleation kinetics as described by Jaroslav Nyvlt [36]. Therefore, HA crystals could grow faster than they nucleate at low BCP coating solution supersaturations, resulting in larger crystal sizes. Higher supersaturations, however, would result in small crystals, since nucleation would be favored compared to their growth [36]. The visibility of Ti6Al4V peaks in diffractograms can be substantiated by CaP layers with a lower thickness than the penetration depth of CuKα radiation into the sample. The atomic molar Ca/P ratio of stoichiometric HA is 1.67 [44]. However, CO_3_^2−^, Mg^2+^, and Na^+^ ions can substitute Ca^2+^-ions in hydroxyapatite, which correlates with non-stoichiometric biological bone tissue [72]. This substitution could explain the reduced amounts of calcium and therefore comparatively low Ca/P ratios of ≈1.5 in the coatings [58,73]. These Ca/P ratios resembled those of natural calcium-deficient apatite [56]. CaP coatings could therefore be described as bone-like HA. BCPx2 featured a Ca/P ratio of 1.67. This Ca/P increase could be explained with an increased substitution of PO_3_^2−^ with CO_3_^2−^ following a B-type substitution as described by Müller et al. [44]. They suggested that HPO_4_^2−^ of calcium-deficient apatite, Ca_10-x_(HPO_4_)_x_(PO_4_)_6-x_(OH)_2-x_, could be substituted by CO_3_^2−^, leading to an increased Ca/P ratio. However, considering the quantitative EDX results, it was found that it was not the carbon content of the detected surface elements that was increased upon increased BCP coating solution ion concentrations, but the calcium. This could be explained by the fact that in contrast to the work of Müller et al. [44], not only the carbonate content of the SBF was increased, but each component of the immersion solution (i.e., also calcium). Hence, HPO_4_^2-^ could be easier substituted with Ca^2+^, which could improve the formation of HA without calcium deficiency.

### 4.2. In Vitro Biocompatibility of CaP-Coatings

Primary human osteoblasts were cultured on the biomimetic coatings precipitated on chemically pre-treated Ti6Al4V for biocompatibility testing. The lowest amount of cell DNA was detected from cells grown on pure Ti6Al4V. All CaP coatings led to a higher level of DNA and therefore, to a higher cell proliferation on these surfaces. HA coatings are known to improve the cell attachment and proliferation of bone marrow-derived mesenchymal stem cells (BMSC) and osteoblast-like cells [16,17,18]. Likewise, the amorphous HA coatings used in this study had a beneficial impact on the osteoblastic expansion, confirming the results of Rydén et al. [74]. Cells seeded on amorphous HA adhered in a higher cell number compared to Ti and crystalline HA after 24 h [74]. In addition, Müller et al. compared amorphous versus crystalline calcium phosphate and revealed a significantly higher viability of cells grown on the amorphous structure [75]. Due to the similarities of amorphous coatings with bone tissue surfaces, osteoblasts and osteoblasts-related-cells could be positively influenced on their attachment possibilities and their proliferation [69,70]. Collagen I is an important protein of the extracellular matrix and therefore essential for bone formation processes [76]. Different studies have shown the increased expression and release of collagen I when osteoblasts and osteoblast-related cells were cultured on HA compared to titanium [16,18]. The coating BCPx1.5 led to the highest type I C-terminal collagen propeptide (CICP) concentration in the supernatant. BCPx1.5, compared to the other surfaces, had the lowest average roughness and was most homogeneous through the enhanced crystal growth compared to BCPx1, but lower crystal growth compared to CaBCPx1.5 and BCPx2. It can be assumed that the homogenous HA surface stimulates collagen I synthesis. This is supported by the lower concentration of CICP measured for the rougher Ti6Al4V and BCPx1 and the even rougher CaBCPx1.5 and BCPx2. CaBCPx1.5 with the highest average roughness and heterogenous Ca structure led to the lowest measured CICP concentration. Therefore, the roughness and structure of the HA surface are important for osteoblastic proliferation and differentiation and an ideal surface texture can influence bone remodeling processes in vitro [77]. Osteoblasts released more collagen I when cultured on hydroxyapatite surfaces with a lower roughness compared to higher roughness [18,78]. As similar observations could be made in this study, the positive influence of a homogeneous CaP surface on osteoblastic differentiation can be assumed. However, this requires further studies to investigate the influence of appropriate surfaces on osteogenic differentiation over a longer period of time. In addition to early osteogenic markers, the suitability of the CaP coating for late differentiation and mineralization should be analyzed.

The inflammatory response to biomaterials is closely linked to osseous tissue regeneration at both the cellular and molecular levels [79]. IL-6 and IL-8 are known to act as a proinflammatory and influence bone resorption processes through an enhanced activation of osteoclast, leading to a reduction of extracellular matrix in bone [25,80,81,82,83,84]. On the other hand, IL-6 can not only reduce bone formation, but can also be necessary for the differentiation of stem cells into osteoblasts and plays an important role in the regulation of osteogenesis [27,28,29]. As IL-6 and IL-8 are important markers for the biocompatibility of materials and their surface modifications, we determined the release of both cytokines as well as VEGF by human osteoblasts. Jämsen et al. revealed, among others, the correlation between the presence of IL-6 and IL-8 in the tissue around hip implants after aseptic loosening [9]. A high level of IL-6 was connected to a similar increase in IL-8, while an early IL-8 expression correlated with an early time of revision of hip implants [22]. In this study, the released cytokine concentration of IL-6 resembled the concentration of IL-8 relating to the different surfaces. Unmodified Ti6Al4V led to a higher concentration of IL-6 as well as a high concentration of IL-8. Modifications with different BCP coating solution concentrations, especially BCPx2 and CaBCPx1.5, reduced the amount of IL-6 and IL-8, suggesting that the used calcium phosphate coatings enhance the biocompatibility through the reduction of proinflammatory chemokines. A similar reduction of IL-6 was noticed by Aniket et al. [85]. The coating with a silica-calcium phosphate nanocomposite reduced the processed IL-6 amount of osteoblasts compared to pure Ti6Al4V. Nicoletti et al. revealed an influence of HA on osteoblasts by decreasing the amount of released IL-6 when incorporated in a demineralized bone matrix [86]. In contrast to IL-8, which has a documented negative effect on osteogenesis [22,29,87], a certain concentration of IL-6 is important to induce osteoblastic differentiation [28]. In this study, surface modifications with BCPx1.5, CaBCPx1.5, and BCPx2 showed a significant reduction in IL-8, implementing improved biocompatibility, but only BCPx1.5 had no significant reduction in IL-6. The IL-6 concentration for BCPx1.5 was, compared to CaBCPx1.5 and BCPx2, just slightly downregulated in relation to pure Ti6Al4V. This could lead to a positive reduction of inflammatory responses and peri-implant diseases [23,25,80,81,82,83,84], but also would leave enough signaling for the differentiation of osteoblasts and osteoblast-like cells [27,28,29]. VEGF is another important maker for inflammatory and early angiogenic response as well as for the vascular development and therefore for successful bone repair processes [88,89,90]. The optimal amount of VEGF is thereby critical for optimal bone regeneration. Besides the positive effects, an local oversupply with VEGF can lead to an inhibition of osteoblast maturation, reduced infiltration of regenerative cells, and activation of osteoclasts [90,91]. The uncoated Ti6Al4V surfaces in our experiment led to a high concentration of IL-6, IL-8, and VEGF, resulting in strong signals for inflammation and infiltration of immune cells like monocytes or lymphocytes [88,89,92]. This inflammatory environment would induce bone remodeling processes leading to an inhibited initial activity of osteoblasts. Through the BCP-coatings, the inflammatory response is reduced, giving osteoblasts the chance for proliferation and differentiation in the first 48 h. Long-term studies are necessary to obtain an overview over the changes in the inflammatory response as they are important for bone remodeling and vascular development on long-term basis. Further studies should be done to improve the understanding of how the roughness and crystallinity of HA surfaces can improve biocompatibility or even positively influence bone remodeling processes. Additionally, the cultivation with different cell types like osteoclasts or endothelial cells can provide more information about bone remodeling processes and the influence on the vascularization on CaP modified implant surfaces.

## 5. Conclusions

In this study, biomimetic CaP coatings were generated on Ti6Al4V implant substrates. Bone-like apatite was formed on chemically pre-treated substrates after immersion in modified BCP coating solutions for 14 days. BCP coating solution modification via supersaturations (1.5x, 2x) and CaCl_2_ pretreatment was discovered to permit modifiable surface roughness, crystallinity, crystal size, and composition of the CaP coating. Due to the coatings, an increased cell proliferation as well as a change in the CICP, IL-6, IL-8, and VEGF concentration could be detected. Despite the preliminary results, BCPx1.5 could be declared to generate the most promising biomimetic coating. Using BCPx1.5 in this cost-effective, low-temperature coating process, homogeneous, smooth, and nanocrystalline bonelike HA coatings were created, leading to the most promising biocompatibility with the highest release of type I collagen propeptide, a significant decrease in IL-8 concentration, and a slight reduction in IL-6 and VEGF after 48 h. Further profound experiments regarding the mechanical characterization of the coating (i.e., adhesion strength), the degradation behavior, long-term cell differentiation, and osseointegration are required.

## Figures and Tables

**Figure 1 materials-14-03516-f001:**
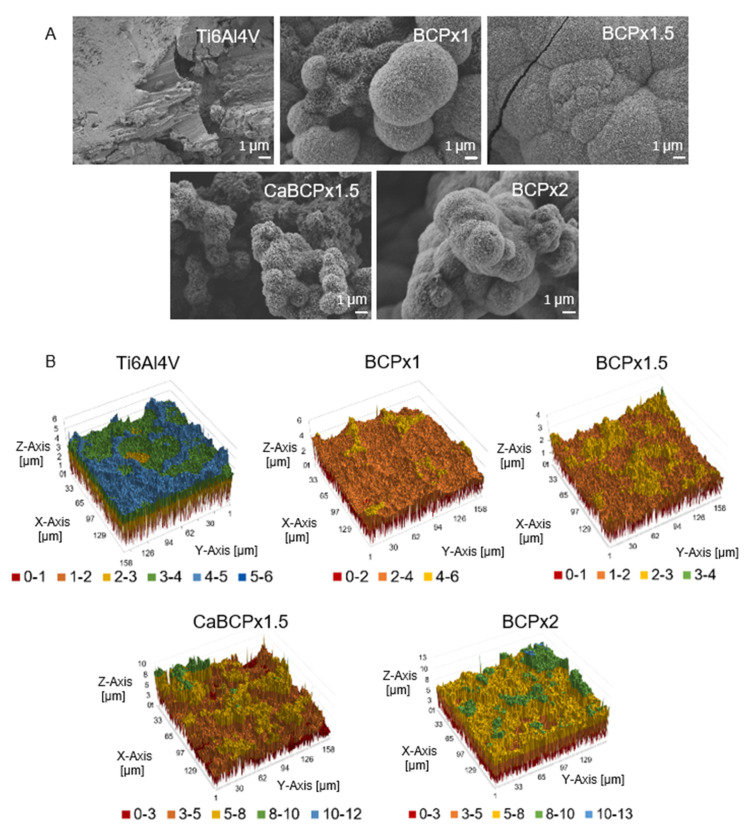
Analysis of biomimetic coatings precipitated on variable chemically pre-treated Ti6Al4V substrates after exposure to BCPx1, BCPx1.5, and BCPx2 for 14 days. (**A**) SEM analysis, Scale: 1 µm (overview images in Appendix A). (**B**) Laserprofilometry scans of the different surfaces.

**Figure 2 materials-14-03516-f002:**
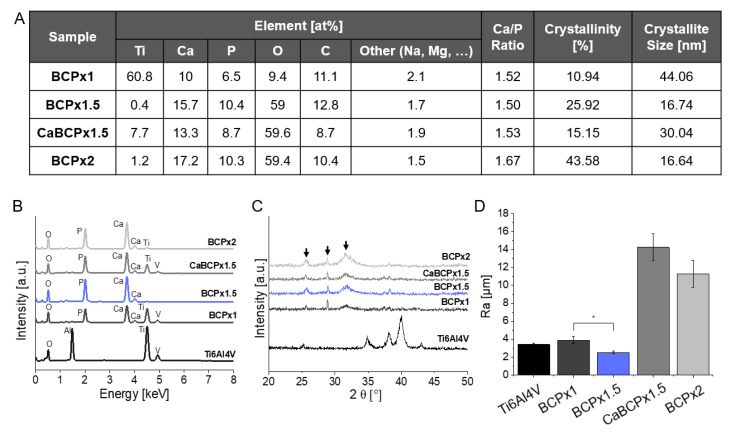
Analysis of biomimetic coatings precipitated on chemically pre-treated Ti6Al4V substrates after exposure to BCPx1, BCPx1.5 with and without CaCl_2_ pre-treatment, and BCPx2 for 14 days. (**A**) Quantitative EDX elemental analysis, crystallinity, and crystallite sizes of examined surfaces. (**B**) Qualitative EDX elemental analysis. (**C**) XRD analysis. Arrows indicate HCA-related peak positions. (**D**) Arithmetic average roughness R_a_. The asterisks (*) indicate statistically significant differences using the Tukey test with a significance level of *p* < 0.05.

**Figure 3 materials-14-03516-f003:**
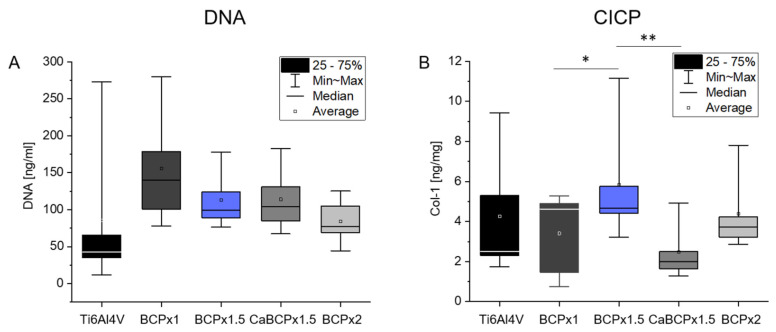
Analysis of biocompatibility from coatings precipitated on chemically pre-treated Ti6Al4V substrates after exposure to BCPx1, BCPx1.5 with and without CaCl2 pre-treatment, and BCPx2 for 14 days using human osteoblasts. (*n* = 5). (**A**) DNA concentration in ng/mL of lysed cells. (**B**) Type I C-Terminal Collagen Propeptide (CICP) concentration in the cell supernatant normalized to the total protein content of the supernatant in ng/mg. * indicates significant differences between CaP coatings: * *p* < 0.05 and ** *p* < 0.01.

**Figure 4 materials-14-03516-f004:**
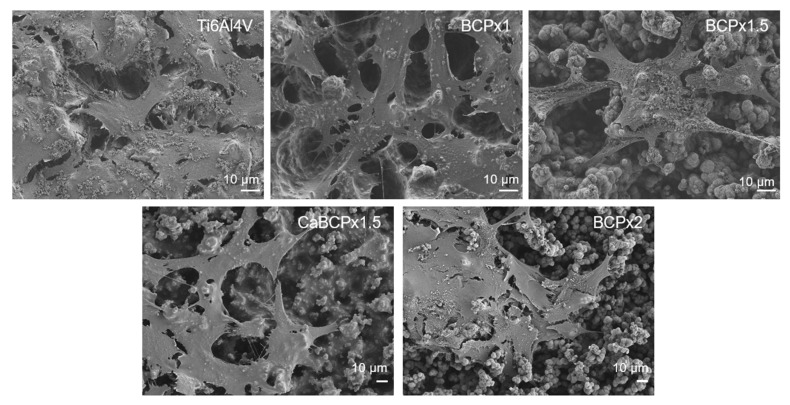
Cell morphology of human osteoblasts growing on coatings precipitated on chemically pre-treated Ti6Al4V substrates after exposure to BCPx1, BCPx1.5 with and without CaCl2 pre-treatment, and BCPx2 for 14 days. Images were taken 48 h after cell seeding with an SEM, scale: 10 µm. Overview images can be found in the Appendix A.

**Figure 5 materials-14-03516-f005:**
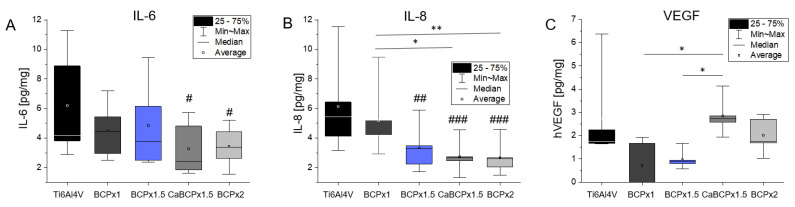
Analysis of biocompatibility from coatings precipitated on chemically pre-treated Ti6Al4V substrates after exposure to BCPx1, BCPx1.5 with and without CaCl2 pre-treatment, and BCPx2 for 14 days using human osteoblasts (*n* = 5). (**A**) Interleukine 6 (IL-6) concentration in the cell supernatant normalized to the total protein content of the supernatant in pg/mg. (**B**) Interleukine 8 (IL-8) in the cell supernatant concentration normalized to the total protein content of the supernatant in pg/mg. (**C**) Vascular Endothelial Growth Factor (VEGF) in the cell supernatant concentration normalized to the total protein content of the supernatant in pg/mg. # indicates significant differences between the CaP coatings and Ti6Al4V: ^#^ *p* < 0.05, ^##^ *p* < 0.01, and ^###^ *p* < 0.001; * indicates significant differences between CaP coatings: * *p* < 0.05 and ** *p* < 0.01.

**Table 1 materials-14-03516-t001:** Ion concentrations and pH value of modified biomimetic calcium phosphate (BCP) coating solutions in comparison to blood plasma.

	Ion Concentrations [mM]			
	Blood Plasma	BCPx1	BCPx1.5	BCPx2
Na^+^	142.0	142.0	213.0	284.0
K^+^	5.0	5.0	7.5	10.0
Mg^2+^	1.5	1.5	2.25	3.0
Ca^2+^	2.5	1.99	2.98	3.98
Cl^−^	103.0	147.8	221.7	295.6
HCO^3−^	27.0	4.2	6.3	8.4
HPO_4_^2−^	1.0	1.0	1.5	2.0
SO_4_^2−^	0.5	0.5	0.75	1.0
pH	7.2–7.4	7.4	7.4	7.4

## Data Availability

The data presented in this study are available on request from the corresponding author.

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
