# Peer review of "Biomimetic Calcium Phosphate Coatings for Bioactivation of Titanium Implant Surfaces: Methodological Approach and In Vitro Evaluation of Biocompatibility"

_materials, 2021, doi:10.3390/ma14133516_

Round 1

Reviewer 1 Report

This work aimed to investigate how to tailor biomimetic CaP coatings on Ti6Al4V substrates using modified biomimetic calcium phosphate (BCP) coating solutions. Undoubtedly, the work is innovative, raises only some reservations when it comes to the very interpretation of the results. 
The statistical reliability is satisfactory. All experiments regarding the structure and composition of CaP-coatings were performed using a minimum of n = 3 replicates. Biocompatibility tests used n = 5 replicates. Normality tests and analysis of variance homogeneity were performed using the Shapiro-Wilk test. Statistically significant differences between means were determined at a value of p < 0.05 as determined by the Tukey posthoc test using a one-way analysis of variances (ANOVA) for normal distribution and the Friedmann analysis for not normal distribution. What is interesting, however, is the modest number of passages of cultivation under standard cell culture conditions.
Looking at Figures 2B and 2C I wonder what was the reason the different definition of intensity units [cps] and [a. u.]. I strongly suggest standardizing it. The other intriguing aspect concerns preferred crystal growth orientations of HCA crystals (Fig. 2C). According to the Authors' deduction decreased Ra values of biomimetic calcium phosphate (BCPx1.5) samples could be explained by enhanced crystal growth due to BCP coating solution supersaturation and preferred crystal growth orientations of HCA crystals in direction of their c-axis. The last-mentioned conclusion is in contradiction with the diffraction data where the (220) is the narrowest peak. If I remember correctly, hydroxyl carbonated apatite does not crystallize in an orthogonal system. That point needs an explanation.
In the quantitative EDX results, it was found that it was not the carbon content of the detected surface elements that was increased upon increased BCP coating solution ion concentrations but the calcium. What is, according to the authors' knowledge, the stabilization mechanism of such an immersion solution?
The BCPx1.5 coating led to the highest CICP concentration in the supernatant (Figure 3B). At the same time, BCPx1.5 has the lowest average roughness and is the most uniform one. On the other hand, the concentration of CICP decreases for the other systems. In addition, osteoblasts released more collagen I when grown on less rough hydroxyapatite surfaces. Thus, the influence of the CaP surface on osteoblastic differentiation needs to be elucidated.

Author Response

Dear Reviewer 1,

we thank the reviewers for their valuable and constructive feedback. We have carefully revised the manuscript and marked the minor correction in yellow in the revised version. The edits are addressed in detail in the responses to the reviewer’s comments below. We strongly believe that the comments have significantly improved the manuscript and hope that it is suitable for publication in its present form.

Yours Sincerely,

Rainer Detsch

-------------------

Reviewer #1:

This work aimed to investigate how to tailor biomimetic CaP coatings on Ti6Al4V substrates using modified biomimetic calcium phosphate (BCP) coating solutions. Undoubtedly, the work is innovative, raises only some reservations when it comes to the very interpretation of the results.

The statistical reliability is satisfactory. All experiments regarding the structure and composition of CaP-coatings were performed using a minimum of n = 3 replicates. Biocompatibility tests used n = 5 replicates. Normality tests and analysis of variance homogeneity were performed using the Shapiro-Wilk test. Statistically significant differences between means were determined at a value of p < 0.05 as determined by the Tukey posthoc test using a one-way analysis of variances (ANOVA) for normal distribution and the Friedmann analysis for not normal distribution.

  1. What is interesting, however, is the modest number of passages of cultivation under standard cell culture conditions.

Our answer: We thank the reviewer for this helpful remark and we improved our manuscript:

For the experiments cells from five different donors, 3 females (age: 75 ± 10 years) and 2 males (age: 60 ± 7 years), were thawed, cultured for another passage, and then seeded on six CaP-coated Ti6Al4V and six untreated Ti6Al4V slides with a density of 22637 cells/cm² in a 12 well plate (passage 4).

  1. Looking at Figures 2B and 2C I wonder what was the reason the different definition of intensity units [cps] and [a. u.].

Our answer: Thank you very much for this comment. Since both unit types can be converted into each other without any problems, we adapted the figure for reasons of consistency.

  1. The other intriguing aspect concerns the preferred crystal growth orientations of HCA crystals (Fig. 2C). According to the Authors' deduction decreased Ra values of biomimetic calcium phosphate (BCPx1.5) samples could be explained by enhanced crystal growth due to BCP coating solution supersaturation and preferred crystal growth orientations of HCA crystals in direction of their c-axis. The last-mentioned conclusion is in contradiction with the diffraction data where the (220) is the narrowest peak. If I remember correctly, hydroxyl carbonated apatite does not crystallize in an orthogonal system. That point needs an explanation.

Answer: We agree with the reviewers comment: Hydroxyapatite crystallizes hexagonally with a preferred c-axis orientation of the deposited crystals [1,2]. We, therefore, verified the crystal planes and corrected the crystal plane (2 2 0) to the (2 1 0) plane:

p.8, lines 256-257: “… hydroxyl carbonated apatite (HCA) related peaks at 2θ = 25,78° (0 0 2), 2θ = 28,9° (2 1 0), and 2θ = 31.74° (2 1 1) …”p.12, line 409: “…(HCA) related peaks at 2θ = 25,78° (0 0 2), 2θ = 28,9° (2 1 0), and 2θ = 31.74° (2 1 1) …”

  1. In the quantitative EDX results, it was found that it was not the carbon content of the detected surface elements that was increased upon increased BCP coating solution ion concentrations but the calcium. What is, according to the authors' knowledge, the stabilization mechanism of such an immersion solution?

Our answer: We thank the reviewer for this important highlight. The stability of the coating solution depends on thermodynamic principles. Following the principle of Gibbs free energy, the dissolution processes of the salts can only take place in an energetically favorable manner if the generation of a higher-order state is possible. This can be achieved through the formation of “molecular aggregates” from water molecules and crystal building blocks. The ions surrounded by water molecules represent a higher-order state compared to that of the crystal lattice. If the salt is not soluble, for example, if there are not enough water molecules to surround ions, the solution is said to be saturated and the salt remains as sediment. In this state, we speak of a dynamic chemical equilibrium in which as much salt precipitates as it dissolves again. In this work, the further we increased the number of ions, the closer we moved to the saturation limit. This became particularly visible when the coating solutions are stored for several weeks (4-6 weeks). In stronger concentrated solutions (i.e., BCPx2) partially precipitated salts can be visible in this time frame. However, all solutions shown in this work are stable at least over the coating time of the Ti6Al4V substrates. The temperature also plays an important role. Rising temperatures cause the water molecules to move faster and the particles of the crystal lattice to vibrate stronger. This allows the lattice energy, i.e., the interaction between the crystal components that hold the crystal together, to be overcome more quickly. Finally, the pH needs to be considered since the concentration of H3O+ and OH- ions influences the reaction quotient which directly affects the Gibbs Free Energy change which determines whether reactions like the solubility of ions will happen spontaneous or not [3].

  1. The BCPx1.5 coating led to the highest CICP concentration in the supernatant (Figure 3B). At the same time, BCPx1.5 has the lowest average roughness and is the most uniform one. On the other hand, the concentration of CICP decreases for the other systems. In addition, osteoblasts released more collagen I when grown on less rough hydroxyapatite surfaces. Thus, the influence of the CaP surface on osteoblastic differentiation needs to be elucidated.

Our answer: We absolutely completely agree with the reviewer. The examination of the osteoblastic differentiation is an important part to validate the CaP coatings. However, for the experimental setting within the present work, we focused on biocompatibility testing of human osteoblasts. To determine differentiation, longer incubation times are necessary to validate late osteogenesis and mineralization which will be part of our next research. To further emphasize your valid input the chapter “4.2 In vitro Biocompatibility of CaP-Coatings” now reads:

However, this requires further studies to investigate the influence of appropriate sur-faces on osteogenic differentiation over a longer period of time. In addition to early osteogenic markers, the suitability of the CaP coating for late differentiation and min-eralization should be analyzed. (p.13 to 14, lines 470-475):”

However, this requires further studies to investigate the influence of appropriate surfaces on osteogenic differentiation over a longer period of time. In addition to early osteogenic markers, the suitability of the CaP coating for late differentiation and mineralization should be analyzed.”

Reviewer 2 Report

The manuscript entitled " Biomimetic Calcium Phosphate Coatings for Bioactivation of Titanium Implant Surfaces: Methodological Approach and In vitro Evaluation of Biocompatibility"  deals with the investigation how to tailor biomimetic CaP coatings on Ti6Al4V substrates using modified biomimetic calcium phosphate (BCP) coating solutions.

The content is very interesting and is worth to publish in Materials.

However, before acceptance, some changes should be performed.

Please revise the manuscript according to the following comments:

Key words: should be in alphabetical order, KEY WORDS should not contain the same words that are within the title of the text.  Thus these should be changed appropriately

Introduction:

Add a new reference to the statement “Among metals,titanium and its alloys are considered as the gold standard due to their biocompatibility, corrosion resistance, and their lack of harmful elements, such as nickel, cobalt, and chromium.”doi: 10.17219/acem/62456

This statement is not a truth „However, titanium and its alloys are often encapsulated by fibrous tissue after implantation due to limited ingrowth of the surrounding tissue [2,6,7].“

M&M

How many samples/titanium discs were used in the study. Describe groups in the study and how many samples were prepared per group.

Author Response

Dear Reviewer 2,

we thank the reviewers for their valuable and constructive feedback. We have carefully revised the manuscript and marked the minor correction in yellow in the revised version. The edits are addressed in detail in the responses to the reviewer’s comments below. We strongly believe that the comments have significantly improved the manuscript and hope that it is suitable for publication in its present form.

Yours Sincerely,

Rainer Detsch

-----------------------

Reviewer #2:

The manuscript entitled "Biomimetic Calcium Phosphate Coatings for Bioactivation of Titanium Implant Surfaces: Methodological Approach and In-vitro Evaluation of Biocompatibility" deals with the investigation of how to tailor biomimetic CaP coatings on Ti6Al4V substrates using modified biomimetic calcium phosphate (BCP) coating solutions.

The content is very interesting and is worth publishing in Materials. However, before acceptance, some changes should be performed. Please revise the manuscript according to the following comments:

  1. Keywords: should be in alphabetical order, KEYWORDS should not contain the same words that are within the title of the text. Thus, these should be changed appropriately.

Our answer: We thank the reviewer for the positive feedback and valuable remarks on the manuscript. The Keywords were adapted. The keywords now read (p.1, line 26): “Hydroxyapatite, Interleukins, Primary human osteoblasts, Ti6Al4V”.

  1. Add a new reference to the statement “Among metals, titanium and its alloys are considered as the gold standard due to their biocompatibility, corrosion resistance, and their lack of harmful elements, such as nickel, cobalt, and chromium.” doi: 10.17219/acem/62456

Our answer: Thank you for pointing this out. We added a new reference in the introduction (Calin et al, DOI: https://doi.org/10.1016/j.msec.2012.11.015).

  1. This statement is not a truth „However, titanium and its alloys are often encapsulated by fibrous tissue after implantation due to limited ingrowth of the surrounding tissue [2,6,7]. “

Our answer: We would like to clarify our statement. We did not want to present an absolute truth but a possibility of the encapsulation of unfunctionalized and dense titanium implants due to weak interfacial bonds between implant surface and bone tissue. To account for the reviewer's important input, we specified this statement. The introduction now reads (p.1, lines 39-41):

“However, dense titanium and its alloys can be encapsulated by fibrous tissue after implantation due to limited ingrowth of bone and weak interfacial bonds to the surrounding tissue [3,7,8].”

  1. How many samples/titanium discs were used in the study. Describe groups in the study and how many samples were prepared per group.

Our answer: We agree and thank the reviewer for the important remark. The chapter “2.2 Surface analysis” was changed into (p.4, lines 121-122):”

Six Ti6Al4V plates were prepared of each sample type (Ti6Al4V, BCPx1, BCPx1.5, BCPx2 and CaBCPx1.5) for surface analysis.“

For the chapter “2.3 Cell culture of primary human osteoblasts” we wrote (p.4, lines 158-161):

"For the experiments cells from five different donors, 3 females (age: 75 ± 10 years) and 2 males (age: 60 ± 7 years), were thawed, cultured for another passage, and then seeded on six CaP-coated Ti6Al4V and six untreated Ti6Al4V slides with a density of 22637 cells/cm² in a 12 well plate.“

Reviewer 3 Report

Executive Summary

The manuscript titled “Biomimetic Calcium Phosphate Coatings for Bioactivation of Titanium Implant Surfaces: Methodological Approach and In vitro Evaluation of Biocompatibility” investigated coating technology and related influence on Titanium implant for medical use. Overall, the research is well designed and scientifically written. The authors may perform minor revisions to further improve the quality of the article.

Major Comments

  • The authors are giving duo intents to IL-6, indicating that IL-6 is a biomarker for inflammation, as well as bone recovery. It would be helpful if the authors can discuss which direction is more important in this research so readers will not be confused.
  • The authors may contact the editorial office to seek some proofreading suggestions.
  • Line 131: the supplementary data was not uploaded so I do not have access to it.

Minor Comments

  • In all figures, please provide full names for shortcuts. For example, CICP in figure 3.
  • Figure 4: please provide more qualification descriptions for the change of morphology, even though there was no negative impact.

Author Response

Dear Reviewer 3,

we thank the reviewers for their valuable and constructive feedback. We have carefully revised the manuscript and marked the minor correction in yellow in the revised version. The edits are addressed in detail in the responses to the reviewer’s comments below. We strongly believe that the comments have significantly improved the manuscript and hope that it is suitable for publication in its present form.

Yours Sincerely,

Rainer Detsch

------------------

Reviewer #3:

The manuscript titled “Biomimetic Calcium Phosphate Coatings for Bioactivation of Titanium Implant Surfaces: Methodological Approach and in vitro Evaluation of Biocompatibility” investigated coating technology and related influence on Titanium implant for medical use. Overall, the research is well designed and scientifically written. The authors may perform minor revisions to further improve the quality of the article.

  1. The authors are giving duo intents to IL-6, indicating that IL-6 is a biomarker for inflammation, as well as bone recovery. It would be helpful if the authors can discuss which direction is more important in this research so readers will not be confused.

Our answer: We thank the reviewer for the important highlight. We clarified this issue in the chapter “4.2 In vitro Biocompatibility of CaP-Coatings”(p.14, lines 475-483):” The inflammatory response to biomaterials is closely linked to osseous tissue regeneration at both the cellular and molecular levels [4] . IL-6 and IL-8 are known to act proinflammatory and influence bone resorption processes through an enhanced activation of osteoclast leading to a reduction of extracellular matrix in bone [5–10]. On the other hand, IL-6 can not only reduce bone formation but also can be necessary for the differentiation of stem cells into osteoblasts and plays an important role in the regulation of osteogenesis [11–13]. As IL-6 and IL-8 are important markers for the biocompatibility of materials and their surface modifications, we determined the release of both cytokines as well as VEGF by human osteoblasts. … (lines 497-498) … In contrast to IL-8, which has a documented negative effect on osteogenesis [13–15], a certain concentration of IL-6 is important to induce osteoblastic differentiation [12].”

Additionally, we rephrased the section 3.2 (p.10, line 312) into “Induction of Cytokine Release by human osteoblasts” to avoid confusion.

  1. The authors may contact the editorial office to seek some proofreading suggestions.

Our answer: Thank you for the advice and if our manuscript is accepted by the reviewers and editors for publication, we will check everything very carefully again.

  1. Line 131: the supplementary data was not uploaded so I do not have access to it.

Our Answer; Answer: We thank the reviewer for this information. It seems there was afailure occurring during the upload of the edited manuscript. We apologize for this and please find the chapter “Supplementary Materials” attached to the manuscript and not uploaded in an extra file.

  1. In all figures, please provide full names for shortcuts. For example, CICP in figure 3.

Our answer: We thank the reviewer for this valid point. We added the full names in each figure description. Figure description 3 (p.9): …

“B: Type I C-Terminal Collagen Propeptide (CICP) concentration in the cell supernatant normalized to the total protein content of the supernatant in ng/mg. * indicates significant differences between CaP coatings: * p < 0.05 and ** p < 0.01.”

Figure description 5 (p.10):” Analysis of biocompatibility from coatings precipitated on chemically pre-treated Ti6Al4V substrates after exposure to BCPx1, BCPx1.5 with and without CaCl2 pre-treatment, and BCPx2 for 14 days using human osteoblasts (n=5). A: Interleukine 6 (IL-6) concentration in the cell supernatant normalized to the total protein content of the supernatant in pg/mg B: Interleukine 8 (IL-8) in the cell supernatant concentration normalized to the total protein content of the supernatant in pg/mg. C: Vascular Endothelial Growth Factor (VEGF) in the cell supernatant concentration normalized to the total protein content of the supernatant in pg/mg. # indicates significant differences between the CaP coatings and Ti6Al4V: # p < 0.05, ## p < 0.01 and ### p < 0.001; * indicates significant differences between CaP coatings: * p < 0.05 and ** p < 0.01.”

  1. Figure 4: please provide more qualification descriptions for the change of morphology, even though there was no negative impact.

Our Answer: Thank you very much for this comment. We revised our Results to better highlight the cell morphology. The chapter “3.2 In vitro Biocompatibility of CaP-Coatings” now reads (p.9, lines 300-305):

“No significant differences on cell morphology of the CaP structures could be noticed. On all CaP surfaces, osteoblasts extensively span the structures. In contrast to the titanium surface and the BCPx2 coating, osteoblasts on BCPx1, BCPx1.5 and CaBCPx1.5 appear to form more numerous filopodia. This is also evident in the lower magnification of the overview images (Fig. 7).”

Literature:

[1] F.A. Müller, L. Müller, D. Caillard, E. Conforto, Preferred growth orientation of biomimetic apatite crystals, J. Cryst. Growth. 304 (2007) 464–471. https://doi.org/10.1016/j.jcrysgro.2007.03.014.

[2] L. Müller, F.A. Müller, Preparation of SBF with different HCO3- content and its influence on the composition of biomimetic apatites, Acta Biomater. 2 (2006) 181–189. https://doi.org/10.1016/j.actbio.2005.11.001.

[3] A. Lüttge, Crystal dissolution kinetics and Gibbs free energy, J. Electron Spectros. Relat. Phenomena. 150 (2006) 248–259. https://doi.org/10.1016/j.elspec.2005.06.007.